# Characterization of the organic vs. inorganic fraction of suspended particulate matter in coastal waters based on ocean color radiometry remote sensing

Hubert Loisel[1], Lucile Duforêt-Gaurier[1], Trung Kien Tran[1], Daniel Schaffer Ferreira Jorge[1], François Steinmetz[2], Antoine Mangin[3], Marine Bretagnon[3], and Odile Hembise Fanton d'Andon[3]

[1]Université du Littoral Côte d'Opale, CNRS, Univ. Lille, UMR 8187 – LOG – Laboratoire d'Océanologie et de Géosciences, F-62930 Wimereux, France
[2]HYGEOS, Euratechnologies, 165 avenue de Bretagne, 59000 Lille, France
[3]ACRI-ST, 260 Route du Pin Montard, 06904 Sophia-Antipolis, France

**Correspondence:** LOISEL Hubert (hubert.loisel@univ-litoral.fr)

**Abstract.** Knowledge of the organic and inorganic particulate fraction of suspended material in coastal waters is essential for the study of particle dynamics and biogeochemical cycles in these complex and highly variable environments. Thanks to the availability of appropriate spatial sensors, and to the considerable improvements of algorithms dedicated to the satellite observation of coastal waters from ocean color radiometry (OCR) achieved these two last decades, various optical and biogeochemical parameters can now routinely be monitored over coastal waters. Here we show that a Proxy of Particulate Composition (PPC) can be estimated from OCR observations. The present algorithm, based on Neural Network approach, has been validated using a broad range of biogeochemical data collected in various contrasted coastal waters, and applied to MERIS observations over the global coastal ocean at a 1km×1km spatial resolution from 2002 to 2012. The relevance of the temporal occurrence of PPC at a given water pixel has been illustrated over the global coastal ocean, and its pertinence has been deeply discussed over the English Channel and southern North Sea which are characterized by a well-documented variability of suspended particulate matter composition. The present algorithm can directly be applied to all OCR sensors.

*Copyright statement.* TEXT

## 1 Introduction

Water quality parameters display large spatio-temporal variability in coastal waters as these areas are the location of strong coupling between aquatic and terrestrial systems and are under the pressure of large natural and anthropogenic forcing. The characterization of the spatio-temporal distribution of biological, biogeochemical, and physical parameters in coastal waters is of a fundamental importance for a variety of applications dedicated to coastal management, which often contains economic interests, and to improve our understanding of the dynamics of the coastal ecosystems and their associated biogeochemical cycles.

Among the large set of water quality parameters sampled in coastal waters, the concentration of the suspended particulate matter (SPM), or equivalent parameters (i.e. the turbidity), has been intensively studied by the scientific community as a key parameter to understand the sediment transport, downstream sedimentation, and coastal geo-morphological processes (Velegrakis et al., 1997; Lahet and Stramski, 2010; Vantrepotte et al., 2012; Loisel et al., 2014; Anthony et al., 2015; Marchesiello et al., 2019). For instance, combining SPM spatio-temporal products with waves data allows resuspension areas to be identified (Loisel et al., 2014). While the SPM spatio-temporal patterns provide relevant information on the suspended particulate dynamics, the variation of the composition (i.e. chemical nature) of SPM may disclose relevant information on the complex chemical, physical, and biological processes occurring in coastal waters. For instance, the portion of particulate organic matter (POM) of SPM is of particular interest when investigating adsorption of trace elements on particles, abrupt changes of water quality due to pelagic or benthic blooms, or fate of suspended matter between the water column and the sediments. Modeling the particulate transport requires specific modules coupled with hydrodynamical models. For example, in the SUBSTANCE module (Mengual et al., 2017), which can be coupled with CROCO, a French code built upon the well-known Regional Oceanic Modeling System (Shchepetkin and McWilliams, 2005), conservative and non-conservative (i.e. biological) particulate substances has to be defined with their own density and/or settling velocity, depending on the SPM nature (mineral vs. organic). The settling and flocculation processes, tightly linked to the organic fraction of SPM, to the transport of particulate variables in water column are also considered in these modules.

The particulate organic carbon (POC), encompasses living (phytoplankton, heterotrophic bacteria, and virus) and non-living (i.e. detritus) organic particles in suspension. Information on the SPM and POC variability represents crucial inputs for the initiation and validation of sediment transport models (Douillet et al., 2001; Ford and Fox, 2014; Wu et al., 2020) and biogeochemical models (Aumont et al., 2015). While the analysis of the variation of the absolute values of SPM and POC concentrations bring relevant information for our understanding of carbon cycle and marine particulate dynamics in coastal waters, the dimensionless POC/SPM ratio is used in many studies to describe the temporal variability in the particulate matter pool composition and origin (Coynel et al., 2005; Emmerton et al., 2008; Doxaran et al., 2012a, 2015; Ehn et al., 2019), which is often related to variation in the regional environmental forcing (e.g. water discharge, phytoplankton bloom dynamics), but also to better interpret the in situ optical measurements (Babin et al., 2003a; Loisel et al., 2007; Woźniak et al., 2011; Doxaran et al., 2012b; Neukermans et al., 2016; Reynolds et al., 2016; Reynolds and Stramski, 2019) and satellite ocean color radiometry (OCR) observations (Vantrepotte et al., 2011). From in situ data collected in the near-shore marine environment at Imperial Beach in California, Woźniak et al. (2010) set threshold values of the POC/SPM ratio to identify changes in the particulate assemblage from the dominance of mineral particles (POC/SPM<0.06) to the dominance of organic particles (POC/SPM >0.25), and mixed particulate assemblages (0.25< POC/SPM <0.6). Neukermans et al. (2016) and Reynolds et al. (2016) followed the approach of Woźniak et al. (2010) to partition their Arctic seawater datasets into these three broad compositional classes.

Due to the high variability of the physical and biogeochemical processes occurring in coastal waters, traditional approaches such as oceanographic cruises and in situ time series, although essential, are very time-consuming, expensive and sometimes uncertain to yield meaningful results on the studied phenomena. Satellite observation of OCR is now well recognized as a powerful tool to monitor the spatio-temporal distribution of biogeochemical and optical parameters in coastal waters (IOCCG,

2000; Loisel et al., 2013; IOCCG, 2018; Groom et al., 2019). Over the past decades, various OCR bio-optical algorithms have shown that SPM (Ahn et al., 2006; Nechad et al., 2010; Feng et al., 2014; Han et al., 2016; Balasubramanian et al., 2020; Pahlevan et al., 2022) and POC concentrations (Liu et al., 2015; Hu et al., 2016; Woźniak et al., 2016; Le et al., 2017; Tran et al., 2019) can be estimated over coastal waters.

The objective of this study is to provide a proxy of particulate composition (PPC) from remote sensing. For that purpose,
the POC/SPM ratio value is used as an intermediate product which allows the PPC, composed by three different classes (i.e. organic-dominated, mineral-dominated, and mix), to be estimated. We will re-examine the relevance of the POC/SPM threshold values of Woźniak et al. (2010), developed to assess the organic vs. mineral fraction of the bulk particulate matter. This will be done through the examination of the relationship between the POC/SPM ratio and the $b_{bp}/c_p$ optical ratio which is an indicator of the bulk particulate assemblage chemical composition (Twardowski et al., 2001; Loisel et al., 2007). An extensive in situ
data set collected in bio-geochemical contrasted environment has been gathered for that purpose. A Neural Network algorithm will be developed and validated against a large in situ data set collected in various coastal environments to assess PPC from OCR. The new algorithm will then be applied to the MERIS observations (2002-2012) over the global coastal waters at 1 km$^2$ of spatial resolution to discuss the significance of this new product. A specific focus will be done on the English Channel and southern North Sea for which the temporal occurrence of the organic vs. inorganic fraction at a given pixel, built from 10-year
temporal series, will be discussed.

## 2 Materials and methods

### 2.1 Dataset description

#### 2.1.1 In situ datasets

In situ measurements were collected between 1997 and 2014 in southeastern Beaufort Sea (Bélanger et al., 2008), French
Guiana (Vantrepotte et al., 2012, 2015), European coastal waters (Baltic Sea, English Channel, North Sea, Bay of Biscay) (Babin et al., 2003a, b; Lubac and Loisel, 2007; Lubac et al., 2008; Neukermans et al., 2012; Bonato et al., 2016; Novoa et al., 2017), and Vietnam East Sea (Loisel et al., 2014, 2017). The sampling strategies, the field measurement protocols and the data processing are described in related papers. Measurements include concentrations of POC and SPM ($\mu$g l$^{-1}$), remote-sensing reflectances ($R_{rs}$, sr$^{-1}$), particulate backscattering coefficient ($b_{bp}$), and particulate attenuation coefficient ($c_p$) (m$^{-1}$) at 650
nm. The samples cover a wide range of biogeochemical variability as POC and SPM concentrations span 4 and 3 orders of magnitude, respectively. The first in situ database, named DS0, includes 300 coincident POC, SPM, $b_{bp}$, and $c_p$ measurements (Table 1). In section 2.3, DS0 will be used to examine the empirical relationship between $b_{bp}/c_p$ and POC/SPM. The POC/SPM ratio ranges between $1.1 \times 10^{-3}$ and $8.7 \times 10^{-1}$. In comparison, Woźniak et al. (2010) collected 44 samples in Imperial Beach near-shore (California) with POC/SPM values between $2.3 \times 10^{-2}$ and $4.2 \times 10^{-1}$. The second in situ database, referenced as
DS, is made of 325 coincident POC, SPM, and $R_{rs}$ measurements with $3.9 \times 10^{-4} \leq$ POC/SPM $\leq 5.6 \times 10^{-1}$ (Table 1). It will be used to test the performance of POC/SPM estimates from the neural network algorithm (section 3.1).

**Table 1.** Information on the in situ data used in this study: mean, [minimum, maximum], and standard deviation (StdDev)

| Dataset | Region | Nb. data | Years | In situ $\frac{POC}{SPM}$ | $\frac{POC}{SPM}$ derived from satellite $R_{rs}$ | $\frac{POC}{SPM}$ derived from in situ $R_{rs}$ | $\frac{b_{bp}(650)}{c_p(550)}$ |
|---|---|---|---|---|---|---|---|
| DS0 | European coastal waters | 300 | 2010-2014 | 0.1136 [0.0001,0.8700] (0.1465) | - | - | 0.0162 [0.0013-0.519] (0.0076) |
| DS | European coastal waters French Guiana, Beaufort Sea Vietnam East Sea | 325 | 1997-2018 | 0.0895 [0.0004,0.5606] (0.1046) | - | 0.0894 [0.0026-0.5382] (0.1024) | - |
| DSM | French coastal waters European coastal waters | 101 | 2002-2018 | 0.0801 [<0.0001,0.5606] (0.1074) | 0.1116 [0.0002,0.5574] (0.1117) | - | - |

### 2.1.2 The global coastal MERIS $R_{rs}$ and Match-up data sets

MERIS level 1 data were used to study the PPC spatial and temporal distribution (Sect. 3.2) and for the match-up exercise (Sect. 3.1). MERIS level 1 data ($\sim$ 1 km pixel resolution) over the 2002-2012 period were processed using the polymer atmospheric correction algorithm (Steinmetz et al., 2011; Steinmetz and Ramon, 2018), which was adapted for coastal waters in the frame of the GlobCoast project. Following Mélin and Vantrepotte (2015), only pixels presenting a distance to the coast lower than 200 km and with a bottom depth not deeper than 4000 m are selected (Loisel et al., 2017). A third dataset, named DSM (for DataSet Match-up), was composed of collocated MERIS data with in situ data points of POC and SPM (Table 1), collected in the framework of the French Coastal Monitoring Network SOMLIT (Service d'Observation en Milieu Littoral, https://www.somlit.fr). The criteria considered for the match-up selection are described in (Bailey and Werdell, 2006). Due to the absence of organic-dominated match-up data points using the MERIS sensor, complementary match-up data points were added to DSM by looking at SeaWiFS match-up with DS. We kept only the match-up data points with a good $R_{rs}$ retrieval (only possible using DS). For that purpose, only data points with $R_{rs}$(in situ)/$R_{rs}$(satellite) values, from 412 to 560 nm, below 0.5 or above 1.5 are selected. The DSM dataset is composed of 101 matched points after the application of these criteria. The POC/SPM mean value is of 0.0801 for DSM instead of 0.1136 and 0.0895, for DS0 and DS, respectively.

### 2.2 Algorithm development

Two different approaches were initially tested to estimate PPC from $R_{rs}$. For the first approach, SPM is estimated by the Han et al. (2016) 's algorithm (referenced as HA16) which consists of semi-analytical relationships between SPM and $R_{rs}$ in the Red or NIR bands, according to the level of turbidity. Typical band-ratio relationship using red to blue-green bands is used to assess POC from the algorithm of Tran et al. (2019) (referenced as TR19). Once POC and SPM are derived, the POC/SPM ratio is calculated and PPC is estimated using the different threshold values (see section 2.3). However, Tran et al. (2019) showed that POC concentration can be overestimated in presence of mineral waters. It results that the POC/SPM ratio is also overestimated when mineral-dominated waters dominate (not shown). Moreover, the errors on both POC and SPM estimations are additive when the POC/SPM ratio is finally calculated. To limit this error propagation, we focus on the development of a single algorithm to derive in one step the POC/SPM ratio from $R_{rs}$.

For that purpose, a neural network approach has been selected as a second approach. We used a feed-forward network with log-sigmoid hidden neurons and linear output neurons to a coupled Levenberg Marquardt algorithm allowing an efficient back-propagation through the training procedure (Lv et al., 2018; Hagan et al., 1996). The DS dataset was randomly divided into 3 datasets to develop, train, and validate this NN. 60% of the observations were used to construct and train the NN, 20% for its validation and 20% to test its performance independently. The training and validation phases are performed jointly, allowing to stop the training procedure when the generalization of the NN stops improving . We tested several combinations of $R_{rs}$ bands (412, 443, 490, 510, 560 and 665 nm) to best predict the POC/SPM ratio. The final NN architecture was best trained using $R_{rs}$ at 412, 490, 510 and 560 nm as the input layer, two hidden layers (8 and 10 neurons), and one output layer (POC/SPM ratio).

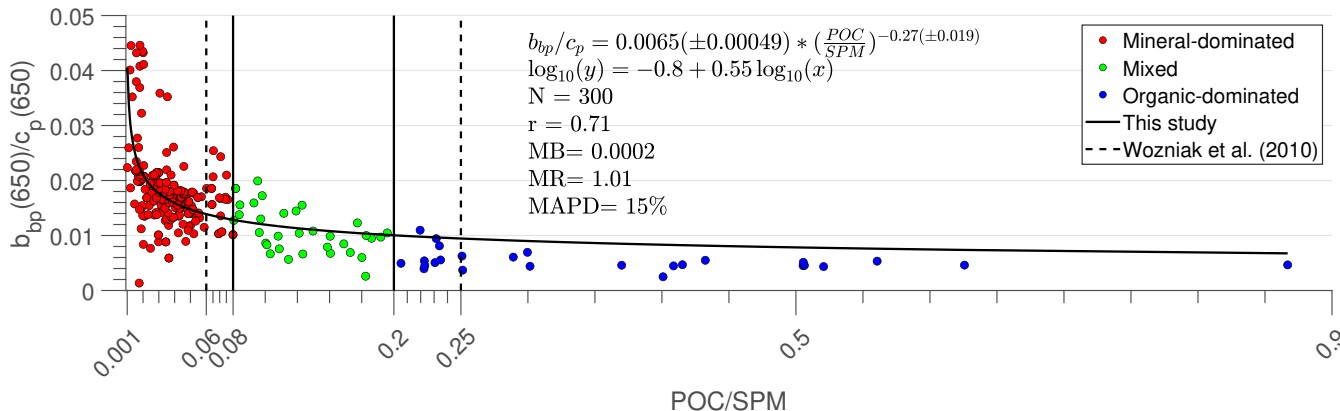

**Figure 1.** Relationships between $b_{bp}/c_p$ (650 nm) and POC/SPM from DS0. The black curve stands for the power regression. The vertical dashed lines represent the thresholds delimiting the water types according to Woźniak et al. (2010). The vertical solid lines show the new thresholds defined in the present study. The linear regression (in log space) between derived (named $y$) and in situ $b_{bp}/c_p$ (named $x$) is written. We wrote also the regression coefficient ($r$), the median absolute percentage difference (MAPD), the median ratio (MR) and the mean bias (MB). These statistical indicators, MAPD, MR and MB, are calculated in normal space as described in Jorge et al. (2021).

The metrics used to evaluate the performance of the NN optimization are described in Portillo Juan and Negro Valdecantos (2022).

### 2.3 Determination of the organic, mineral, and mineral-organic mixed fraction

As previously explained, the POC/SPM ratio is an indicator of the particle assemblage and can be used to partition in situ data into three water types as defined by Woźniak et al. (2010). In addition, some theoretical and field studies showed that the variability of the ratio $b_{bp}/b_p$ can be related to the particle composition (Boss et al., 2004; Twardowski et al., 2001; Loisel et al., 2007; Duforêt-Gaurier et al., 2018). Low $b_{bp}/b_p$ values are observed for a particle population dominated by low refractive index material such as phytoplankton. In contrast, high $b_{bp}/b_p$ are generally observed in the presence of relatively high concentration of inorganic particles. At 650 nm, $c_p$ is dominated by the scattering and the optical ratio $b_{bp}/c_p$ can be used instead of $b_{bp}/b_p$ (Loisel et al., 2007).

The objective here is to re-examine the pertinence of the POC/SPM threshold values of Woźniak et al. (2010) on a larger in situ coastal dataset (DS0) (covering a wider range of optical and biogeochemical variability) through the examination of the POC/SPM versus $b_{bp}/c_p$ relationship. As expected, $b_{bp}/c_p$ decreases when POC/SPM increases, that is when we move from a mineral-dominated to an organic-dominated environment with $b_{bp}/c_p$ values typically lower 0.010-0.012 (Twardowski et al., 2001; Loisel et al., 2007) (Fig. 1). The regression line, $b_{bp}/c_p$ versus POC/SPM (black line), is plotted and the estimated regression coefficients are indicated with their standard error (Fig. 1). The threshold values are first fixed according to the $b_{bp}/c_p$ values (as a given range of $b_{bp}/c_p$ values corresponds to a given range of refractive index of the bulk particulate matter), and then adjusted with a careful examination of each data points for which ancillary data (i.e. chlorophyll-a, counted cells,

phytoplankton to particulate absorption ratio, and $R_{rs}$ spectra) are used to better characterize the bulk particulate matter. The first threshold value has been shifted to 0.08 (corresponding to a $b_{bp}/c_p$ value of 0.012) to encompass data points collected in mineral dominated environments (close to river mouths). An asymptote is reached at high POC/SPM values (above 0.2)

and concerns data points with low $b_{bp}/c_p$ (below 0.075) values typical to phytoplankton dominated environments. The new threshold value of POC/SPM for organic-dominated waters, is set to 0.2 (i.e. $b_{bp}/c_p < 0.075$) to encompass in situ data points collected during bloom events. This value is however very similar to the value previously obtained by Woźniak et al. (2010) (= 0.25). Data points located along the asymptote are therefore associated with organic-dominated waters.

The thresholds are applied to monthly POC/SPM values derived from MERIS data, and the frequency of dominance is

computed for each pixel of the scene over the ten years (2002-2012) as detailed below. A pixel geographically located at a given latitude and longitude is named $k$ pixel (k=1, $S_{tot}$). $S_{tot}$ is the total number of pixels over the selected geographical area. For each $k$ pixel, we computed the number of valid pixels ($N_{tot}^k$) over the period that corresponds to 120 months ($N_{tot}^k \leq$120). The term "valid pixel" means a non-flag pixel for which the $R_{rs}$ value is provided to derive POC/SPM with the neural network. Then, for the $k$ pixel, we calculate how many time the class $i$ is identified over $N_{tot}^k$: $n_i^k$ is the class occurrence, $i$=1, 2 or 3

with 1 for mineral-dominated, 2 for organic-dominated, and 3 for mixed waters. The frequency of dominance is defined as the ratio of the occurrence to the number of valid pixels:

$$D^k = \frac{maximum(n_1^k, n_2^k, n_3^k)}{N_{tot}^k} \times 100(\%) \tag{1}$$

The Equation 1 is repeated $S_{tot}$ times to compute the frequency of dominance over the whole scene and provide maps in section 3.2.

## 3 Results

### 3.1 Validation of the classification

In situ measurements of $R_{rs}$ from DS are used as input to the neural network algorithm to test the performance of the estimation of algorithm-derived values, named NN POC/SPM (Fig. 2a). The NN algorithm achieves a good performance over the whole range of POC/SPM (Fig. 2a). MAPD is 24%, MR is 1.06 and the bias is 0.004. The slope of the type II (log-transformed)

regression is 0.90. The same classification is obtained for 88.5% of data points between the in situ and model derived values. About 10.9% of data point are misclassified in the adjacent group and only 0.62% are misclassified in a non-adjacent group. SeaWiFS and MERIS match-up on both the SOMLIT in situ independent dataset (for which no in situ $R_{rs}$ are available) and DS in situ data (for which in situ $R_{rs}$ are available) are shown in Fig. 2b. Compared to the results obtained using in situ measurements (Fig. 2a), the data points are much more scattered around the 1:1 line when POC/SPM is derived from satellite

$R_{rs}$ (Fig. 2b). This strongly emphasizes the impact of atmospheric correction on PPC which is a common feature of OCR products. In Figure 2b, all mineral-dominated data are from the SOMLIT dataset. As the in situ $R_{rs}$ are not available for SOMLIT data, we cannot apply the criteria described in section 2.1.2 to remove inaccurate satellite $R_{rs}$ retrievals. Removing such $R_{rs}$ will certainly allow us to identify strong misclassified patterns. In Fig. 2b, 68.4 % of mineral-dominated data, 44.4

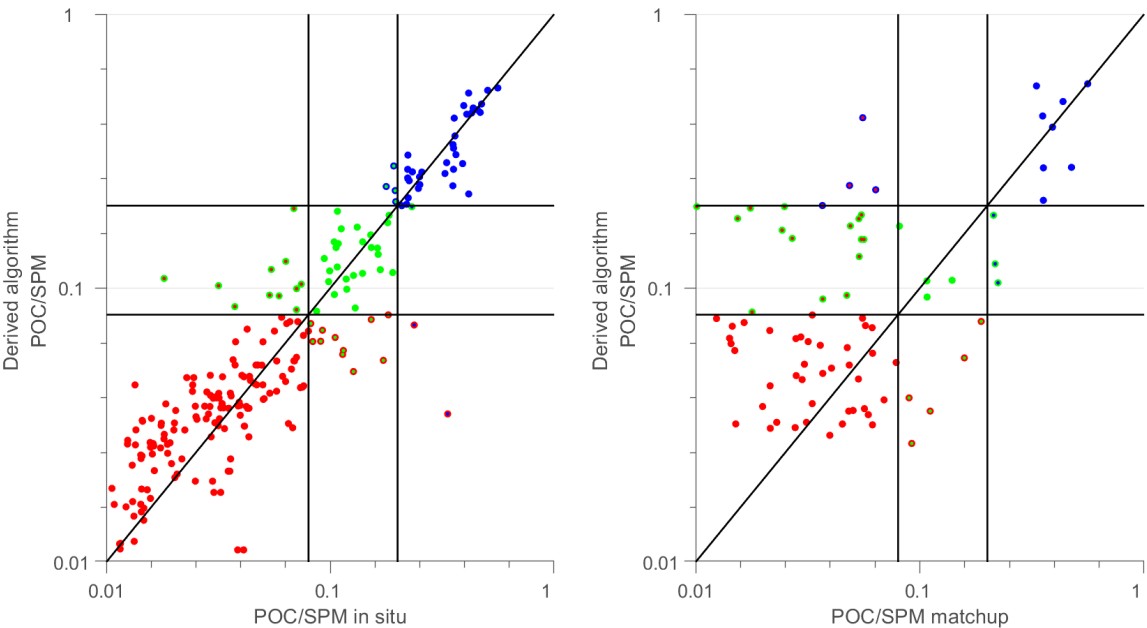

**Figure 2.** Comparison of in situ and derived POC/SPM from neural network algorithm (NN POC/SPM) (a) for DS (i.e., in situ $R_{rs}$ are used by NN POC/SPM) (b) for DSM (i.e., satellite $R_{rs}$ are used by NN POC/SPM). The dashed line is the 1:1 line. Red, blue and green color are for mineral-dominated, organic-dominated and mixed waters, respectively. For both panels, the vertical and horizontal solid lines indicate the thresholds (=0.08 and 0.2) used to partition the water types. Dots are color-labeled according to the values of the in situ POC/SPM, whereas the circles are color-labeled according to the values of POC/SPM derived from NN. When circles and dots are of the same color that means that the retrieved values belong to the same class as the in situ ones.

% of mixed data, and 73.7 % of organic-dominated data are well-classified. Due to atmospheric correction uncertainties, a

proper estimation of POC/SPM values from remote sensing is still very challenging, while the estimation of PPC can still been performed with a reasonable accuracy. Extra match-up data points, including both in situ $R_{rs}$ and POC/SPM measurements, are however needed to definitely support this conclusion.

## 3.2    Global and regional PPC patterns

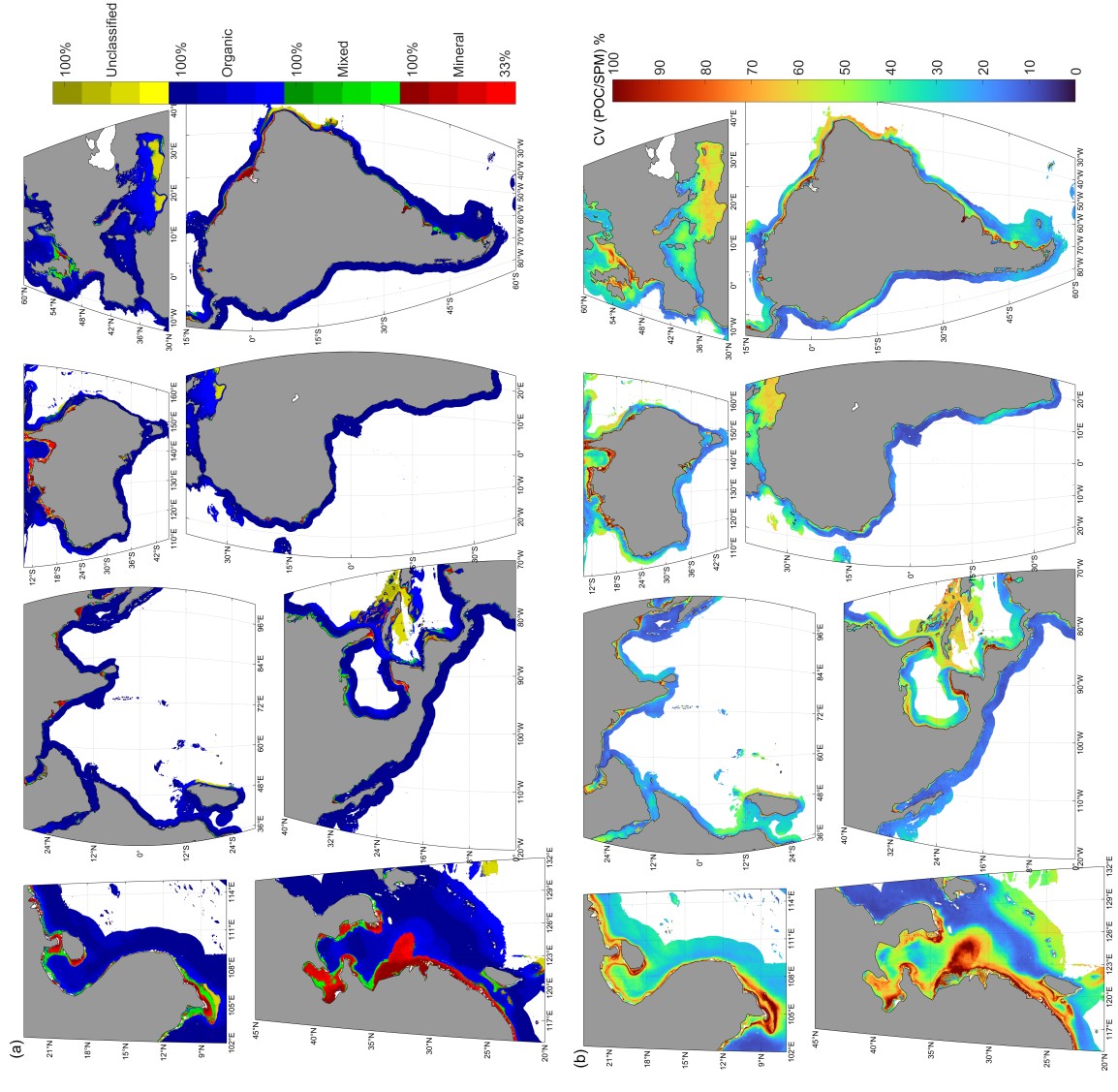

**Figure 3.** (a) Frequency (%) of dominant water-type (red for mineral-dominated/green for mixed/ blue for organic-dominated) over the MERIS time period 2002-2012. The yellow color corresponds to unclassified pixels that means pixels for which the retrieved value of POC/SPM is non-valid (i.e., negative) (b) Coefficient of variation of POC/SPM over the MERIS period 2002-2012

The PPC algorithm has been applied to MERIS monthly $R_{rs}(\lambda)$ data over the 2002-2012 time period, and the maximum of occurrence of PPC has been calculated over this 10-years time period as described in section 2.3 (Fig. 3a). As expected, we observe a well-marked near-shore off-shore gradient from mineral-dominated waters to organic-dominated waters with a transition zone corresponding to mixed particulate assemblages. Large deltas and estuaries characterized by intense sediment discharge such as the Amazon estuary, Mekong delta, Huanghe (Yellow River) delta, Yangtze estuary, or Ganges Brahmaputra delta, present very high occurrence of mineral-dominated waters (70-90%). These areas can extend much further from the river outlet, depending on the bathymetry and the surface currents, as for instance in the central part of the Yellow Sea in front of the Yangtze estuary. Values of POC/SPM of 0.12-0.15, depicting mix-dominated particulate assemblages have been measured in the Eastern China Sea region (124-126°E; 30-32°N) in great coherence with our present finding (Hung et al., 2007). The dominance of mineral dominated waters is generally persistent all over the considered time period as depicted by the relatively low (20-30%) coefficient of variation observed at the outlet of these different estuaries and deltas (Fig. 3b). In contrast, in front of the Amazon and Yangtze estuaries and Huanghe delta some off-shore areas of high (70%) coefficient of variation depict the impact of seasonal and inter-annual interactions between the continental (river and sediment discharges) and oceanic forcing (tide, wind, surface current, sub-mesoscale structures) and the bathymetry. For example, tight interaction between the Amazon plume dominated by mineral particles and the retroflection of the North Brazil Current carrying oceanic particulate organic matter occur in region in front of the Amazon estuary where the slope in the continental shelf drastically changes (Gensac et al., 2016; Varona et al., 2019), causing a significant temporal variation of the PPC. In contrast to the previously mentioned rivers, and despite the Congo river is the world's second largest river in terms both of drainage area and water discharge, only a small PPC signature (mainly organic-dominated) signature can be noticed at the outlet of its estuary (not visible in Fig. 3a), except in the coefficient of variation (Fig. 3b). This is coherent with the fact that the sediment discharge of the Congo river is very low, in contrast to its dissolved part, and that most of the particulate assemble is organic (Coynel et al., 2005). A specific zoom on this area reveal the dominance of organic matter even at the outlet of the estuary, in good agreement with in situ data reported in Coynel et al. (2005).

While an extensive investigation of the different spatio-temporal patterns presented in Fig. 3 are out of the scope of the present paper, a specific focus illustrating the pertinence of the PPC product is conducted over the eastern English Channel (EEC) and southern North Sea (SNS) for which the biogeochemical and physical processes driving the suspended particulate matter composition are relatively well-documented. The eastern English Channel and southern North Sea coastal region is known for its strong hydrodynamics marked by intense tidal currents. This shallow water region, therefore largely impacted by resuspension effects, is also largely influenced by freshwater discharges from different rivers, la Seine and la Somme being the most important ones along the French coast of EEC, and Thames and Scheldt along the SNS. Strong spring and early summer phytoplankton blooms of diatoms and Phaeocystis globosa occurred in this region at different time of the year, depending on the nutrient and light (i.e. season and turbidity level) availability (Breton et al., 2000; Antajan et al., 2004; Sazhin et al., 2007). For these different reasons, the composition of the suspended particulate matter in this area is strongly variable, as already stressed from different field studies based on optical and biogeochemical measurements (Loisel et al., 2007; Vantrepotte et al., 2007; Lubac et al., 2008). The monthly occurrence of each PPC class is in excellent agreement with our present knowledge

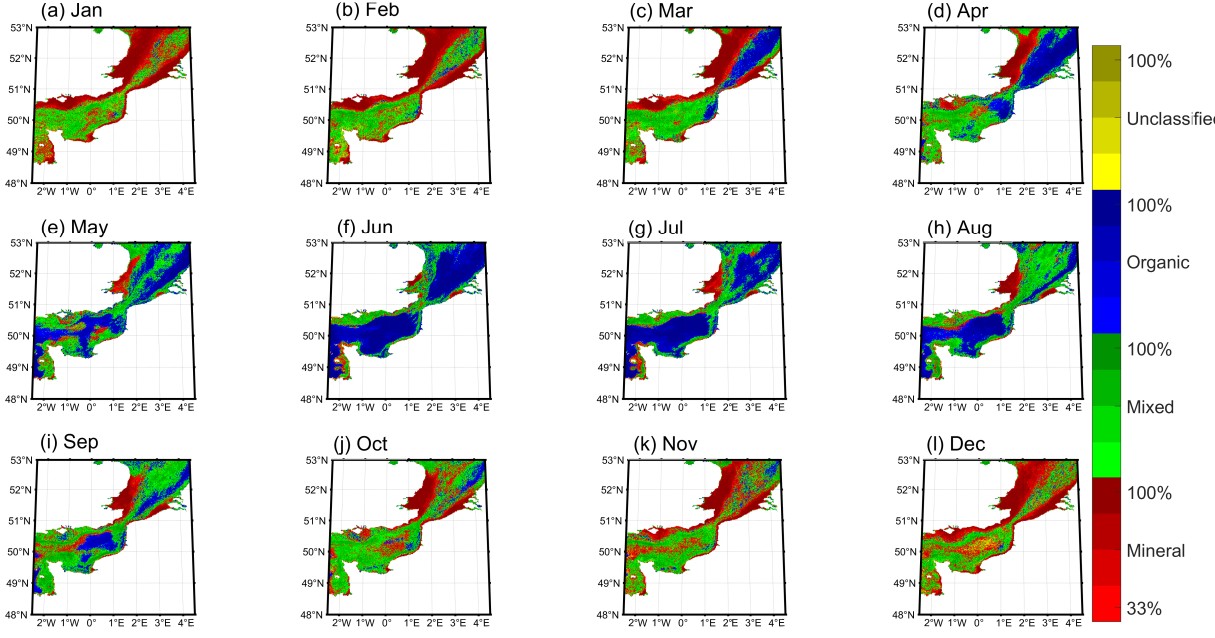

**Figure 4.** Frequency of dominance (%) for mineral-dominated (red), mixed (green,) organic-dominated waters (blue), or unclassified (yellow) over the MERIS period 2002-2012 in the eastern English Channel and southern North Sea.

of this area (Fig. 4). Between mid-fall and winter (from October to February), the particulate matter pool of EEC and SNS
is characterized by mineral and mixed-dominated assemblages mainly due to sediment river inputs and resuspension effects due to strong physical (i.e. waves) forcing (Lesourd et al., 2003; Lancelot et al., 2007; Gohin, 2011; Renosh et al., 2014). This is consistent with results of Vantrepotte et al. (2012) who observed that relative contribution of particulate matter assemblage with a strong proportion of mineral particles increases significantly during the winter time period. In March, organic-dominated pixels appear in the SNS coastal region and in front of the Somme river area in the EEC. Then, while the organic dominated
particles assemblages areas are developing in SNS from March to July, similar particulate assemblages start developing from May to July in off-shore waters of EEC, and along the EEC coast from June. The timing of the development of organic matter areas in the EEC and SNS is in perfect agreement with phytoplankton blooms dynamics in these regions which first develop in the Bay of Somme area and SNS followed by another bloom which developed southern in the Seine river bay area (Brunet et al., 1996; Vantrepotte et al., 2007; Joint and Pomroy, 1993; Van der Zande et al., 2012; Gohin et al., 2019). Finally, areas
covered by organic-dominated pixels are maximum in June/July, and then mix assemblages start developing in the northern part of the domain to fully cover the whole region in October. Mix assemblages are more likely dominated by aggregates of mineral particles and detrital matter (non-living organic particles) which developed during the senescence of the bloom (Vantrepotte et al., 2007; Loisel et al., 2007; Gohin, 2011; Grattepanche et al., 2011).

## 4  Concluding remarks

The PPC can be used as a proper indicator for characterizing the nature of the bulk suspended particulate matter from satellite OCR remote sensing. The MERIS spatio-temporal patterns of this new indicator are consistent with general features documented in the literature. Due to visible bands used, common to all OCR sensors, the PPC algorithm can be applied to all OCR satellite sensors allowing to assess the long-term trend of suspended particulate matter over the global coastal ocean. The application of this algorithm to the satellite Copernicus Marine Environment Monitoring Service (CMEMS) data is underway

to assess the different temporal patterns (seasonal, trend, and irregular) using the CENSUS X-11 approach (Vantrepotte et al., 2011).

*Data availability.*  DS0, DS, and DSM can be obtained from the World Data Center PANGAEA (https://www.pangaea.de/?t5Oceans)

*Author contributions.*  H.L. as the major funder proposed the idea of the work, gave permission to data in LOG, was involved in the algorithm development and data analysis, and co-wrote the manuscript. L.D.-G. was involved in the algorithm development, supervised the analysis

and visualization of the data, was involved in manuscript organization and co-writing. T.K.T. analyzed and visualized the in situ and satellite data, developed and tested the algorithms, developed the codes to processed images. D. S. F. J. supported the development of the Matlab processed images, provided the final plots and statistics and, revised the manuscript. F. S. provided Polymer algorithm to perform atmospheric corrections to process satellite images in coastal waters. A. M, M. B. and O. H. F d'A. gave suggestions and comments.

*Competing interests.*  No competing interests are present

*Acknowledgements.*  This study was realized with the support from the French Space Agency (CNES) through the COYOTE project (CNES/-TOSCA program). MERIS data were processed in the framework of the GlobCoast project funded by the French Agence Nationale de la Recherche (ANR 2011 BS56018 01). Some of the in situ measurements used in this study were collected from different cruises performed in Vietnam coastal waters, French Guyana, and Eastern English Channel through various projects funded by CNES. The authors are grateful to all the people that contributed to data provided by the Service d'Observation en Milieu Littoral (SOMLIT), INSU-CNRS. We thank also

our many colleagues, who participated in the collection of various datasets cited in this study. We thanks Arnaud Cauvin and Xavier Mériaux who conducted field missions at LOG, processed field bio-geophysical data and were in charge of LOG database management. We thanks Roy Elhourany for his advice concerning the neural network algorithm. The authors thank the reviewers, who helped to improve the paper with their comments and suggestions.

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
