# Peer review of "Characterization of the organic vs. inorganic fraction of suspended particulate matter in coastal waters based on ocean color radiometry remote sensing"

_State of the Planet, 2022_

## Author Comment (AC1)

The authors thank the reviewer#1 and #2 for the comments and suggestions related to this study. The reviewers' comments are listed below with our corresponding answers.

**Response to reviewer#1**

**Comment#1 -** From a methodology standpoint, this work advances previously published algorithms to assess POC and SPM from ocean color remote sensing to derive POC/SPM via a neural network based algorithm with thresholds used to define different oceanic particle types.

I am not sure I understand the motivation of this work. The authors develop a new algorithm to derive POC/SPM, which would seem to be a significant scientific advance for the community, but the final product is a general classification of water type, i.e., mineral dominated, mixed types, and organic dominated. What is the point exactly of the general classification? Is this to help inform application of other algorithms or global model configurations?

> **Answer -** The interest to provide a classification index from remote sensing was deeply discussed in the introduction of the first version of the paper (lines 26 to 49: investigating adsorption and desorption of trace elements on particles; modeling particulate transport; improved bio-optical algorithms). The objective of this paper is to provide an information on the organic and inorganic particulate fractions of suspended material to the community for the different reasons exposed in the introduction. The POC/SPM ratio value is here just an intermediate product which allows the three classes, used by the community, to be defined. Therefore, more than an accurate POC/SPM ratio retrieval (very difficult to obtain due to atmospheric correction errors), this study focuses on a good estimation of the three different classes.
>
> The following sentence "In this paper, while a POC/SPM algorithm will be developed, a proxy of particulate composition (PPC) will be developed also to classify SPM into its organic, mineral, or mix fractions from OCR space observations"
>
> has been changed into:
>
> "The objective of this study is to provide a proxy of particulate composition (PPC) from remote sensing. For that purpose, the POC/SPM ratio value is used as an intermediate product which allows the PPC, composed by three different classes (i.e. organic dominated, mineral dominated, and mix) to be estimated".

**Comment#2 -** As I mention below, the thresholds for this classification appear arbitrary. The intermediate product of POC/SPM would ostensibly seem more valuable from a science application perspective.

> **Answer -** Concerning the mentioned arbitrary thresholds, they are fixed according to the values of the bbp/cp ratio first, and second based on the use of ancillary data such as the chlorophyll-a concentration, cell counts, and non-algal and phytoplankton absorption coefficients. This optical ratio, has been shown from theoretical and field measurements, to be highly related to the refractive index, that is to the mineral vs. organic fractions, of the

bulk particulate matter (Twardowski et al., 2001; Loisel et al., 2007). Low bbp/cp ratio are related to phytoplankton dominated environments (as explained line 119, in the first version of the paper, line 120-123 in the new version), whereas high bbp/cp are related to environments dominated by minerals.

To better explain this point, we reformulate the previous sentences as follows (Line 61 to 64 in the new version of the paper):

*"We will re-examine the relevance of the POC/SPM threshold values of Wozniak et al. (2010) developed to assess the organic vs. mineral fraction of the bulk particulate matter. This will be done through the examination of the relationship between the POC/SPM ratio and the bbp/cp optical ratio which is an indicator of the bulk particulate assemblage chemical composition (Twardowski et al., 2001; Loisel et al., 2007). An extensive in situ data set collected in bio-geochemical contrasted environment has been gathered for that purpose."*

We also now better explain how ancillary data are used (see answer to comment#6)

**Comment#3 -** The NN POC/SPM algorithm presented in this work appears new, but there are almost no details given. Minimum details needed are: wavebands used, training data set used, and metrics used to determine valid results.

**Answer - Thanks for this comment. A new paragraph with relevant references has now been added to describe the NN algorithm** (line 109 to 118):

*« For that purpose, a neural network approach has been selected. We used a feed-forward network with log-sigmoid hidden neurons and linear output neurons to a coupled Levenberg Marquardt algorithm allowing an efficient back-propagation through the training procedure (Lv et al., 2018; Hagan et al., 1996). The DS dataset was randomly divided into 3 datasets to develop, train, and validate this NN.60% of the observations were used to construct and train the NN, 20% for its validation and 20% to test its performance independently. The training and validation phases are performed jointly, allowing to stop the training procedure when the generalization of the NN stops improving . We tested several combinations of Rrs bands (412, 443, 490, 510, 560 and 665 nm) to best predict the POC/SPM ratio. The final NN architecture was best trained using Rrs at 412, 490, 510 and 560 nm as the input layer, two hidden layers (8 and 10 neurons), and one output layer (POC/SPM ratio). The metrics used to evaluate the performance of the NN optimization are described in Portillo Juan and Negro Valdecantos (2022) »*

**Comment#4 -** In 2.2 it is stated that 2 approaches are tested here, the POC/SPM derived from separate POC and SPM derivations via previously published NN algorithms and the new NN algorithm based on POC/SPM. But results do not specify which algorithm was used for what analyses.

**Answer -** Thanks for this comment, it is now better mentioned line 109 in the new version

of the paper: *"For that purpose, a neural network approach has been selected, and will be used in this study"*

**Comment#5 -** The analysis associated with Fig 1 confused me in the context of study objectives. Thresholds are being set for POC/SPM to specify water types dominated by different particle types, but why is bbp/cp being regressed with POC/SPM? What does bbp/cp add? The objective in exploring these kinds of relationships, as the authors point out, is typically to used bbp/cp as a proxy for POC/SPM. I do not understand how bbp/cp is somehow influencing the decision on where to set thresholds.

**Answer -** This point has now been explained (see answer to comment#2)

**Comment#6 -** Where thresholds are set, in fact, appears arbitrary. The authors state these thresholds are set "Based on our knowledge of the in-situ data points" but what does that mean? Setting of thresholds should be done quantitatively and independently of preconceived expectations in results. On the contrary, it appears the authors varied thresholds until results, i.e., maps of particle types for different regions, fit their preconceived bias. This is not a sound approach in my opinion.

**Answer -** This is now better explained. *"based on our knowledge of in situ data points"* was changed into *"The threshold values are first fixed according to the bbp/cp values (as a given range of bbp/cp values corresponds to a given range of refractive index of the bulk particulate matter), and then adjusted with a careful examination of each data points for which ancillary data (i.e. chlorophyll-a, counted cells, phytoplankton to particulate absorption ratio, and Rrs spectra) are used to better characterize the bulk particulate matter."*

In this paragraph the sentence *"A rapid decrease is observed for POC/SPM values smaller than the first threshold value of Wozniak et al. (2010) (POC/SPM<0.06) corresponding to bbp/cp values higher than 0.013"* was deleted because lead to confusion.

**Comment#7 -** Fig 2b looks like a shot gun target whereas the relationship in Fig 2a looks quite encouraging. In my opinion, 35% misclassified data for particle type in Fig 2b is quite poor. The authors state this is primarily driven by poor atmospheric corrections. Because these are match up data, why can't the in-situ measured Rrs be used to assess algorithm performance independently of the atmospheric correction? In my opinion this should be done. And an important conclusion of the paper should be, apparently, that we need better atmospheric corrections.

**Answer -** Figure 2a already presents the performance, in terms of both POC/SPM and PPC, of the algorithm when only in situ Rrs data are used. We have now added in the original paragraph the reference to Fig 2a and 2 b which were missing, as follows (Line 156 and 160, 164). "In situ measurements of Rrs from DS are used as input to the neural network algorithm to test the performance of the estimation of algorithm-derived values, named NN POC/SPM (Fig. 2a). *"The NN algorithm achieves a good performance over the whole155*

*range of POC/SPM (Fig. 2a). The median absolute percentage difference (MAPD), the median ratio (MR) and the bias are calculated in normal space and are described in Jorge et al. (2021). MAPD is 24%, MR is 1.06 and the bias is 0.004. The slope of the type II (log-transformed) regression is 0.90 »*

and lines 158-160 that *"The same classification is obtained for 88.5% of data points between the in situ and model derived values. About 10.9% of data point are misclassified in the adjacent group and only 0.62% are misclassified in a non-adjacent group."*
Due to atmospheric correction uncertainties, a proper estimation of POC/SPM values are still very challenging, while the estimation of PPC can still been performed with a reasonable accuracy (here 63% of good retrieval of mineral dominated waters).

The sentence *"It means as even if inaccuracies occurs on NN POC/SPM, it does not prevent to obtain a right PPC in most cases"* was replaced by *"Due to atmospheric correction uncertainties, a proper estimation of POC/SPM values from remote sensing is still very challenging, while the estimation of PPC can still been performed with a reasonable accuracy"*.

**Comment#8** - As mentioned above, I would be more interested in performance assessments for the POC/SPM with associated error metrics such as mean absolute % error, mean absolute error, etc, tabulated.

**Answer** - This is provided in the text (MAPD, MR, slope, and bias), and due to the limited allowed figure/table we can not tabulated these values.

**Comment#9** - From the relationship in Fig 2a, it looks like the authors may be on to something! If the authors feel compelled to include a general (arbitrary) classification of particle type as well with global maps, OK, but the performance and applicability of the POC/SPM algorithm should be rigorously documented.

**Answer** - This point has been addressed in the discussion of Fig 2, through the following sentence: *" Due to atmospheric correction uncertainties, a proper estimation of POC/SPM values from remote sensing is still very challenging, while the estimation of PPC can still been performed with a reasonable accuracy"* (lines 165 to 166). A rigorous examination of the propagation of atmospheric correction error, provided by different atmospheric correction algorithms, in the POC/SPM algorithms, would require a full study which is not the scope of the present short paper which focuses on PPC.

**Response to reviewer#2**

**Comment#1** – Line 41: Change to "related"

**Answer –** Done

**Comment#2** – Lines 14-49: This is a monster paragraph. For improved readability, suggest breaking it up into several shorter paragraphs.

> **Answer** : Thehe text is now composed of three paragraphs instead of one.

**Comment#3** – Line 51: Change to "such as"

> **Answer –** Done

**Comment#4 –** Section 2.1.1: Are these data available to the community and, if so, where? Also, a map and a table that more deeply describe the temporal and geophysical ranges for each sub-dataset would both be useful.

> **Answer –** in a first version of the paper, we added a map to describe the geographical location of the data. However, the OSR7 guideline indicate that the length of the section is limited to a maximum of 4 figures. Consequently, we deleted the map.
> We have now stipulated in the acknowledgment that "DS0, DS, and DSM can be obtained from the World Data Center PANGAEA (https://www.pangaea.de/?t5Oceans)."
> A new table describing has been added for that purpose.

**Comment#5 –** Line 80: Was DS0 also used to train the neural network? If not, what dataset was used for this?

> **Answer –** A brief description of the NN approach with the corresponding references for the NN approach were added. We explained that the DS dataset was used to construct, train and validate.
> *« For that purpose, a neural network approach has been selected. We used a feed-forward network with log-sigmoid hidden neurons and linear output neurons to a coupled Levenberg Marquardt algorithm allowing an efficient back-propagation through the training procedure (Lv et al., 2018; Hagan et al., 1996). The DS dataset was randomly divided into 3 datasets to develop, train, and validate this NN.60% of the observations were used to construct and train the NN, 20% for its validation and 20% to test its performance independently. The training and validation phases are performed jointly, allowing to stop the training procedure when the generalization of the NN stops improving . We tested several combinations of Rrs bands (412, 443, 490, 510, 560 and 665 nm) to best predict the POC/SPM ratio. The final NN architecture was best trained using Rrs at 412, 490, 510 and 560 nm as the input layer, two hidden layers (8 and 10 neurons), and one output layer (POC/SPM ratio). The metrics used to evaluate the performance of the NN optimization are described in Portillo Juan and Negro Valdecantos (2022).»*

**Comment#6 -** Line 88: Is this how MERIS data were processed for the full suite of spatial and temporal analyses, or just for the match-up exercise? If the former, please make it clear this section describes ALL MERIS data processing methods.

> **Answer -** MERIS data were processed for the full suite of spatial and temporal analyses and for match-up exercise (as mentioned in the first version of the paper). This is however now better specified.
> We first change the title of section 2.1.2 into "The global coastal MERIS Rrs and Match-up

data sets" and slightly modify the corresponding paragraph as follows:

MERIS level 1 data were used to study the PPC spatial and temporal distribution (Sect. 3.2) and for the match-up exercise (Sect. 3.1). MERIS level 1 data (~ 1 km pixel resolution) over the 2002-2012 period were processed using the polymer atmospheric correction algorithm (Steinmetz et al., 2011; Steinmetz and Ramon, 2018), which was adapted for coastal waters in the frame of the GlobCoast project. Following Mélin and Vantrepotte (2015), only pixels presenting a distance to the coast lower than 200 km and with a bottom depth not deeper than 4000 m are selected (Loisel et al., 2017). A third dataset, named DSM (for DataSet Match-up), was composed of collocated MERIS $R_{rs}$ data with in situ data points of POC and SPM, collected in the framework of the French Coastal Monitoring Network SOMLIT (Service d'Observation en Milieu Littoral, https://www.somlit.fr). The criteria considered for the matchup selection are described in (Bailey and Werdell, 2006). The DSM dataset is composed of 91 matched points after the application of these criteria, and is characterized by lower POC/SPM values than DS0 and DS. The POC/SPM mean value is of 0.0782 for DSM instead of 0.1136 and 0.1043, for DS0 and DS, respectively.

**Comment#7 –** Line 91: Do you mean 400m? If not, does your analysis require any additional explanation re: differences that could be realized in a shallow system where resuspension is meaningful vs. a system off a steep shelf break?

**Answer -** The choice of a broader coastal mask has been adopted to avoid to under-sampling coastal areas. We fully agree that many studies can now been done based on this coastal OCR data set; and coastal areas, as reported here, are not the ones which are only impacted by resuspension effects. The same mask has been used by Mélin and Vantrepotte (2015) for OCR data, but also by space agencies (ESA CMENS products) when coastal products are provided (distance from the coast lower than 200 km). This is now specified in the new version (see answer to comment 5 above)

**Comment #8-** Section 2.1.2: This is a 3rd in situ dataset, correct? If so, as above, additional description would be useful.

**Answer -** We have three dataset. DS0 and DS are described in section 2.1.1 and are exclusively built with in situ data, and a third one, DSM, which is the match-up dataset described in Section 2.1.2.This is now better specify in the new version (see answer to comment 5).

**Comment #9 –** Line 95: Maybe change to "Two different approaches were initially tested to assess PPC."?

**Answer –** Done

**Comment#10 –** Section 2.2: Please consider adding 1-2 sentences each for HA16 and TR19 to briefly describe how these two algorithms operate. Also, please elaborate substantially on the neural network approach and its formulation. Is there a reference for the NN approach?

**Answer –**

A brief description of the NN approach with the corresponding references for the NN approach were added.

*« For that purpose, a neural network approach has been selected . We used a feed-forward network with log-sigmoid hidden neurons and linear output neurons to a coupled Levenberg Marquardt algorithm allowing an efficient back-propagation through the training procedure (Lv et al., 2018; Hagan et al., 1996). The DS dataset was randomly divided into 3 datasets to develop, train, and validate this NN.60% of the observations were used to construct and train the NN, 20% for its validation and 20% to test its performance independently. The training and validation phases are performed jointly, allowing to stop the training procedure when the generalization of the NN stops improving . We tested several combinations of Rrs bands (412, 443, 490, 510, 560 and 665 nm) to best predict the POC/SPM ratio. The final NN architecture was best trained using Rrs at 412, 490, 510 and 560 nm as the input layer, two hidden layers (8 and 10 neurons), and one output layer (POC/SPM ratio). The metrics used to evaluate the performance of the NN optimization are described in Portillo Juan and Negro Valdecantos (2022).»*

*Concerning the description of the algorithms we add the following sentences in the new version of the text:*

*For the first approach, SPM is estimated by the Han et al. (2016) algorithm (referenced as HA16) which consists of semi-analytical relationships between SPM and Rrs the Red or NIR bands, according to the level of turbidity. Typical band-ratio relationship using red to blue-green bands is used to assess POC from the algorithm of Tran et al. (2019) (referenced as TR19).*

**Comment#11 -** Line 108: "correspond"

**Answer –** Done

**Comment#12** – Line 114: Please reword around "that is" or add a comma or semi-colon.

**Answer -** Done

**Comment#13 –** *Lines 116-122: How new thresholds were assigned reads as subjective. Please elaborate on how the in situ information ("bas*ed on our knowledge of the in-situ points") allowed you to define the new thresholds. For example, how did you define "mineral-dominated" in the field data? As mentioned previously, a more complete description of the in situ datasets might help with this.

**Answer -** This is now better explained. *"based on our knowledge of in situ data points"* was changed into *The threshold values are first fixed according to the bbp/cp values (as a given range of bbp/cp values corresponds to a given range of refractive index of the bulk particulate matter), and then adjusted with a careful examination of each data points for which ancillary data (i.e. chlorophyll-a, counted cells, phytoplankton to particulate absorption ratio, and Rrs spectra) are used to better characterize the bulk particulate matter.*

*In this paragraph the sentence "A rapid decrease is observed for POC/SPM values smaller than the first threshold value of Wozniak et al. (2010) (POC/SPM<0.06) corresponding to bbp/cp values higher than 0.013" was deleted because lead to confusion.*

**Comment#14 –** *Line 138: Are median absolute % difference, mean ratio, and slope calculated in log-transform*ed space or normal space? Recommend presenting the equations for the first two.

**Answer –** We add this sentence with a reference : « *The median absolute percentage difference (MAPD), the median ratio (MR) and the bias are calculated in normal space and are described in Jorge et al. (2021).*

Jorge, D. S., Loisel, H., Jamet, C., Dessailly, D., Demaria, J., Bricaud, A., Maritorena, S., Zhang, X., Antoine, D., Kutser, T., Bélanger, S.,Brando, V. O., Werdell, J., Kwiatkowska, E., Mangin, A., and d'Andon, O. F.: A three-step semi analytical algorithm (3SAA) for estimating inherent optical properties over oceanic, coastal, and inland waters from remote sensing reflectance, Remote Sensing of Environment, 263, 112 537, https://doi.org/https://doi.org/10.1016/j.rse.2021.112537, 2021.*(2021) »*

**Comment #15 –** Line 145: Some points are only misclassified by a short distance, whereas others are much farther off the mark. Is there a metric that can be reported to encompass this? Thinking that since the thresholds are somewhat arbitrary (e.g., is there really a difference between 0.08 and 0.09?), you might want to take credit for "near-misses".

**Answer:** We fully agree with your comment, and for the credit of using "near-misses", however such metric would have to be applied for a broader match-up data points (using other OCR sensors as well).

**Comment#16 –** Line 200: Just reiterating that the PPC (neural network) approach needs to be better described within this manuscript.

**Answer –** Done ; please see answer to comment#10.

---

## Author Response (AR2)

**Response to reviewer:** we would like to thank very much the topical editor, Mrs G. Neukermans for the helpful comments.
* * *
*Figures:*
*Fig 1. Basic statistics of the fitted curve are required, including the standard error on the estimated regression coefficients and an RMSE.*

Answer:  Basic statistics were added in Figure 1 and their description is provided in the legend: "*The regression coefficient (r), the median absolute percentage difference (MAPD), the median ratio (MR) and the mean bias (MB) values are indicated. These statistical indicators (MAPD, MR and MB), are calculated in normal space as described in Jorge et al. (2021).*"

The sentence " *The median absolute percentage difference (MAPD), the median ratio (MR) and the bias are calculated in normal space and are described in Jorge et al. (2021).*" in section 3.1 (line 156 in the previous version of the paper) was deleted as it is now in the legend of the figure 1.
* * *
*Fig. 3. Labels for the panels a and b are missing*
*Fig. 4. Labels for all 12 panels are missing; it is not clear which map corresponds to which month.*
*Typos:*
*Line 191: outside the "scope" (instead of "scoop")*

Answer: it was modified.
* * *
*Already pointed out by reviewer 2: line 92 a threshold of 4000 m deep to mask coastal waters seems like an error. Should this be 400 m instead?*

Answer: As we already answer to the reviewer, this is 4000 m, as in the mask defined in Melin and Vantrepotte (2015) to better take into account to some very specific areas where river plumes are observed further than 200 km.
* * *
*The biggest remaining issue is the fact that the paper does not show any evidence that Organic and mixed particle compositions can be (reliably) detected from Ocean color remote sensing (as shown in Fig. 2b); also pointed out by reviewer 1.*
*Why is Fig. 2b limited to SOMLIT data? Why not include the entire in situ dataset of POC and SPM and match that with MERIS reflectance? You might end up with a dataset that includes organic dominated samples.*

Answer: Figure 2b is not limited to SOMLIT data anymore. We added match up obtained from DS dataset (please remind that, the in situ database, referenced as DS, is made of 325 coincident POC, SPM, and $R_{rs}$ measurements, see section 2.1.1). The availability of $R_{rs}$ in situ data points allows us now to discard some in situ POC/SPM match-up data points for which we observe bad satellite $R_{rs}$ retrieval (which is not the case with the original match-up data points coming from only SOMLIT, as $R_{rs}$ are not available in SOMLIT dataset). Figure 2b displays that 73.7% of organic-dominated data points, 44.4% of mixed data points and 68.4% of mineral-dominated data points are well classified. Misclassified data points are due to inaccurate $R_{rs}$ retrievals.

We re-wrote the text from line 162 to line 172.

We added in the text (lines 95-100) "*Due to the absence of organic-dominated match-up data points using the MERIS sensor, complementary match-up data points were added to DSM by looking at SeaWiFS match-up with DS. We kept only the match up data points with a good Rrs retrieval (only possible using DS). For that purpose, only data points with Rrs(in situ)/Rrs(satellite) values, from 412 to 560 nm, below 0.5 or above 1.5 are selected. The DSM dataset is composed of 101 matched points after the application of these criteria. The POC/SPM mean value is of 0.0801 for DSM instead of 0.1136 and 0.0895, for DS0 and DS, respectively.*"

Values concerning DSM were changed in table 1 and lines 99-100 as data points were added in DSM as explained in section 2.1.2.
* * *
*In the light of this remark -Unless I missed something important, I suggest to tone down the conclusion and the abstract which state that PPC can be detected from OCR remote sensing, as there is only evidence that the satellite retrieval works for mineral dominated particle suspensions in 63% of cases, despite the fact that the results presented in Fig. 3 and 4 are encouraging, as well as the results presented in Fig. 2a.*

We showed that misclassified data points are due to inaccurate satellite $R_{rs}$. Using DS, we added match-up for organic and mixed waters. In Fig. 2b, 68.4 % of mineral-dominated data, 44.4 % of mixed data, and 73.7 % of organic-dominated data are well-classified. We showed that misclassified data points are due to inaccurate satellite $R_{rs}$.